# Uric Acid Has Direct Proinflammatory Effects on Human Macrophages by Increasing Proinflammatory Mediators and Bacterial Phagocytosis Probably via URAT1

**DOI:** 10.3390/biom10040576

**Published:** 2020-04-09

**Authors:** Camilo P. Martínez-Reyes, Aarón N. Manjarrez-Reyna, Lucia A. Méndez-García, José A. Aguayo-Guerrero, Beatriz Aguirre-Sierra, Rafael Villalobos-Molina, Yolanda López-Vidal, Karen Bobadilla, Galileo Escobedo

**Affiliations:** 1Laboratory for Proteomics and Metabolomics, Research Division, General Hospital of Mexico “Dr. Eduardo Liceaga”, 06726 Mexico City, Mexico; nava111222@hotmail.com (C.P.M.-R.); aaron.manjarrez@gmail.com (A.N.M.-R.); angelica.mendez.86@hotmail.com (L.A.M.-G.); jose.aguayo01@iest.edu.mx (J.A.A.-G.); bety_agsi@hotmail.com (B.A.-S.); 2Unidad de Biomedicina, Facultad de Estudios Superiores Iztacala, Universidad Nacional Autónoma de México, 54090 Mexico City, Mexico; villalobos@campus.iztacala.unam.mx; 3Programa de Inmunología Molecular Microbiana, Facultad de Medicina, Universidad Nacional Autónoma de México, 04510 Mexico City, Mexico; lvidal@unam.mx; 4Department of Immunology, Instituto Nacional de Enfermedades Respiratorias Ismael Cosío Villegas, 14080 Mexico City, Mexico; karenbolo@hushmail.com

**Keywords:** macrophage, uric acid, TNF-alpha, TLR4, phagocytosis, probenecid, URAT1

## Abstract

The relationship of uric acid with macrophages has not been fully elucidated. We investigated the effect of uric acid on the proinflammatory ability of human macrophages and then examined the possible molecular mechanism involved. Primary human monocytes were differentiated into macrophages for subsequent exposure to 0, 0.23, 0.45, or 0.9 mmol/L uric acid for 12 h, in the presence or absence of 1 mmol/L probenecid. Flow cytometry was used to measure proinflammatory marker production and phagocytic activity that was quantified as a percentage of GFP-labeled *Escherichia coli* positive macrophages. qPCR was used to measure the macrophage expression of the urate anion transporter 1 (URAT1). As compared to control cells, the production of tumor necrosis factor-alpha (TNF-alpha), toll-like receptor 4 (TLR4), and cluster of differentiation (CD) 11c was significantly increased by uric acid. In contrast, macrophages expressing CD206, CX3C-motif chemokine receptor 1 (CX3CR1), and C-C chemokine receptor type 2 (CCR2) were significantly reduced. Uric acid progressively increased macrophage phagocytic activity and downregulated URAT1 expression. Probenecid—a non-specific blocker of URAT1-dependent uric acid transport—inhibited both proinflammatory cytokine production and phagocytic activity in macrophages that were exposed to uric acid. These results suggest that uric acid has direct proinflammatory effects on macrophages possibly via URAT1.

## 1. Introduction

In humans, monosodium urate—also referred to as uric acid—is the end-product of purine metabolism [1]. Uric acid is mainly excreted in urine and then reabsorbed in epithelial cells of the proximal convoluted tubule by urate transporters, such as the urate anion transporter 1 (URAT1), a transmembrane protein highly expressed in endothelial cells, adipocytes, and cartilage chondrocytes [2,3,4,5]. Normal blood levels of uric acid are 2.4–6.0 mg/dL for women and 3.4–7.0 mg/dL for men [6,7]. However, hyperuricemia is a pathological entity that is characterized by blood uric acid values beyond upper limits that has been largely associated with metabolic syndrome and increased cardiovascular risk [8,9,10]. In this sense, a recent study conducted in 22,983 Chinese adults demonstrated that subjects with hyperuricemia had higher prevalence of cardiovascular risk factors, such as dyslipidemia, hypertension, and type 2 diabetes (T2D), than individuals with normal uric acid values [11]. It is also worth mentioning that uric acid concentration associated directly with the number of cardiovascular risk factors; in other words, the cardiovascular risk increases as the uric acid elevates in blood.

The association of hyperuricemia with cardiovascular diseases is accompanied by a low-grade inflammation state that is characterized by increased proinflammatory activation of macrophages [11,12]. In this regard, the exposure of ApoE^−/−^ mice to allopurinol (an inhibitor of uric acid production) does not only reduce atherosclerotic plaque size, but also macrophage infiltration and the expression of tumor necrosis factor-alpha (TNF-alpha) and interleukin (IL-) 1 beta, both typical proinflammatory cytokines [12]. Thus, such a seminal work proposed, for the first time, a relationship between uric acid and macrophages in the scenario of cardiovascular disease.

Macrophages are innate immune cells that differentiate from monocytes and play prominent roles in phagocytosis, inflammation, and wound healing [13]. Macrophages exhibit extremely high plasticity by exerting the ability of orchestrating either proinflammatory responses or anti-inflammatory actions, depending on the extracellular milieu [14]. In the presence of prototypical molecules, such as lipopolysaccharide (LPS) and/or interferon gamma (IFN-gamma), human macrophages show proinflammatory functions by producing TNF-alpha, IL-1 beta, CD11c, and toll-like receptor 4 (TLR4) [15,16]. On the contrary, the exposure of macrophages to dexamethasone or cytokines, such as interleukin IL-4 and IL-13, leads these cells to adopt an anti-inflammatory profile that is characterized by the production of IL-10 and/or CD206 [15]. Furthermore, the exposure of macrophages to pro- or anti-inflammatory stimuli also affects the expression of molecules that are involved in leukocyte trafficking, such as CX3C-motif chemokine receptor 1 (CX3CR1) and C-C chemokine receptor type 2 (CCR2) [17,18]. More importantly, the inflammatory activation enhances the ability of macrophages to ingest bacteria with respect to that described in macrophages with anti-inflammatory functions [19,20].

Although previous evidence suggests that high uric acid levels associate with increased proinflammatory activity of human macrophages, there is not yet conclusive data in this respect. For this reason, we investigated the effect of increasing uric acid concentrations on the proinflammatory or anti-inflammatory ability of primary human macrophages in vitro, and then examined the possible molecular mechanism involved.

## 2. Materials and Methods

### 2.1. Subjects

Human monocytes were isolated from buffy coat suspensions of ten healthy male volunteers aged 18–35 years with no metabolic alterations, attending the Blood Bank of the General Hospital of Mexico. All of the participants provided written informed consent, being previously approved by the institutional ethical committee of the General Hospital of Mexico (registration number of the ethical code approval: DIC/11/UME/05/029), which guaranteed that the study was conducted in rigorous adherence to the principles that were described in the 1964 Declaration of Helsinki and its posterior amendment in 2013. Blood donors were excluded of the study if they had previous diagnosis of type 1 *Diabetes Mellitus* (T1D), T2D, cardiovascular disease, acute or chronic liver disease, acute or chronic renal disease, cancer, endocrine disorders, infectious diseases, and inflammatory or autoimmune disease. We also excluded of the study to HIV, HCV, and HBV-seropositive patients, and subjects with anti-inflammatory, antiplatelet drugs, anti-hypertensive, and immunomodulatory medication, including non-steroidal anti-inflammatory drugs.

### 2.2. Monocyte Isolation and Macrophage Culture

Buffy coat suspensions (*n* = 10) were separately diluted 1:2 with PBS1X (Sigma Aldrich, St. Louis, MO, USA) for the subsequent isolation of peripheral blood mononuclear cells (PBMCs) by density gradient centrifugation while using histopaque-1077 (Sigma Aldrich, St. Louis, MO, USA). The monocytes were then isolated from PBMCs by CD14-negative selection using magnetic columns (Miltenyi Biotec, Bergisch Gladbach, Germany). Purified monocytes were placed in RPMI-1640 medium (Sigma Aldrich, St. Louis, MO, USA) containing 10% fetal bovine serum (FBS) (Gibco^TM^, Grand Island, NY, USA), 2 mM L-glutamine, 50 μg/mL gentamicin, and 10 ng/mL macrophage-colony stimulating factor (M-CSF) (Gibco^TM^, Grand Island, NY, USA) in six-well cell-culture plates (Costar, Kennebunk, ME, USA), at a density of 3 × 10^6^ monocytes per well. Culture media and M-CSF were replaced every other day for seven days. Upon differentiation, the monocyte-derived macrophages (MDM) were cultured in RPMI-1640 medium supplemented, as mentioned above and exposed to 0.23, 0.45, or 0.9 mmol/L uric acid (Sigma Aldrich, St. Louis, MO, USA) for 12 h. After performing several time-response curves at 1, 3, 6, and 12 h, and one, three, six, and nine days, we found that MDM exhibited the most significant proinflammatory activity at 12 h of in vitro culture in the presence of uric acid, and we decided to perform all experiments at this time. Prior uric acid exposure, 1 mmol/L probenecid—a non-specific blocker of URAT1-dependent uric acid transport (St. Louis, MO, USA)—was added and replaced after 6 h by culture media that contained different uric acid concentrations. Control MDM were placed in supplemented RPMI-1640 medium in the absence of uric acid for the same time. Immediately after, MDM were collected while using sterile cell scrapers (Corning, Reynosa, Tamaulipas, Mexico) and equally divided into 1 mL PBS1X (Sigma Aldrich, St. Louis, MO, USA) for flow cytometry analysis or 300 μL TRIzol Reagent (Life Technologies, Carlsbad, CA, USA) for quantitative polymerase chain reaction (qPCR) assays.

### 2.3. Flow Cytometry Assays

After collecting the cells, 1 × 10^6^ MDM were resuspended in 50 μL Cell Staining Buffer (BioLegend, Inc., San Diego, CA, USA). Subsequently, the cells were incubated with anti-CD14 PE/Cy7, anti-CD206 APC/Cy7, anti-TLR4 PE, anti-CX3CR1 BV510, anti-CCR2 AF647 (BioLegend, Inc., San Diego, CA, USA), and 5 μL True-Stain Monocyte Blocker™ (BioLegend, Inc., San Diego, CA, USA) for 20 min. in darkness at 4 °C. After being rinsed with Cell Staining Buffer (BioLegend, Inc., San Diego, CA, USA), MDM were incubated with 7-AAD (BD Pharmingen™, San Jose, CA, USA) for 10 min. for subsequent analysis on a FACSCanto II flow cytometer (BD Biosciences, San Jose, CA, USA) by means of BD FACSDiva^TM^ software 6.0, acquiring 20,000 events per test in triplicate. The compensation controls were performed using UltraComp eBeads™ (Invitrogen, Carlsbad, CA, USA) for each fluorochrome. Flow cytometry data were analyzed using the FlowJo 10.0.7 software (TreeStar, Inc, Ashland, OR, USA). For intracellular cytokine stain, MDM were treated with 1:1000 Brefeldin A (BioLegend, Inc San Diego, CA, USA) for the last 2 h of in vitro culture. After collecting cells, 1 × 10^6^ MDM were resuspended in 50 μL Cell Staining Buffer (BioLegend, Inc., San Diego, CA, USA). Immediately after, the MDM were simultaneously incubated with anti-CD14 PE/Cy7, anti-CD11c PE/Cy5, Zombie UV Fixable Viability Kit, and 5 μL True-Stain Monocyte Blocker™ (BioLegend, Inc., San Diego, CA, USA) for 20 min. in darkness at 4 °C. Afterwards, MDM were incubated with 100 μL Fixation Medium A (FIX & PERM™ Cell Permeabilization Kit) (Invitrogen™, Carlsbad, CA, USA) for 15 min. at room temperature. Afterwards, MDM were rinsed with Cell Staining Buffer (BioLegend, Inc., San Diego, CA, USA) for subsequent incubation with 100 μL Permeabilization Medium B (FIX & PERM™ Cell Permeabilization Kit) (Invitrogen™, Carlsbad, CA, USA), and anti-TNF-alpha AF488 and anti-IL-1 beta Pacific Blue for 20 min. in darkness at room temperature. Immediately after, the MDM were rinsed with Cell Staining Buffer (BioLegend, Inc., San Diego, CA, USA) and then acquired on a BD Influx flow cytometer (BD Biosciences, San Jose, CA, USA) by means of BD Sortware^TM^ software 1.2, acquiring 20,000 events per test in triplicate. Compensation controls were performed using UltraComp eBeads™ (Invitrogen™, Carlsbad, CA, USA) for each fluorochrome. Flow cytometry data were analyzed while using the FlowJo 10.0.7 software (TreeStar, Inc, Ashland, OR, USA).

### 2.4. Macrophage Phagocytic Activity

*Escherichia coli* of the strain DH5a were transformed while using the green fluorescent protein (GFP)-mut2 encoding plasmid pCD353 (*E. coli*-GFP+) that is regulated by a lactac promoter. GFP was induced using 1mM isopropyl-b-D-1-thiogalactopyranoside (Sigma Aldrich, St. Louis, MO, USA), as previously described [21]. *E. coli*-GFP+ were in vitro cultured with macrophages in a ratio of bacteria:cells = 30:1 for 1 h. Afterwards, cell suspensions were centrifuged at 2500 rpm/15 min. to remove remnant bacteria and then resuspended in 50 μL of sterile PBS 1X for posterior analysis on a FACSCanto II flow cytometer (BD Biosciences, San Jose, CA, USA) by means of BD FACSDiva^TM^ software 6.0, acquiring 20,000 events per test in triplicate.

### 2.5. URAT1 Gene Expression by qPCR

After collecting cells, 1 × 10^6^ MDM were resuspended in TRIzol Reagent (Life Technologies, Life Technologies, Carlsbad, CA, USA) at 4 °C. Immediately after, the total RNA was isolated by the phenol/chloroform/guanidine isothiocyanate method according to the manufacturer’s instructions. Total RNA samples were quantified and subjected to reverse transcription using the M-MLV Retrotranscriptase system in the presence of dT primer (Invitrogen, Carlsbad, CA, USA). After being incubated at 37 °C for 60 min., cDNA samples were obtained and used for amplification while using the real-time polymerase chain reaction (qPCR) in the presence of SYBR Green Master Mix and AmpliTaq^®^ Fast DNA Polymerase (Applied biosystems, Foster City, CA, USA), according to the manufacturer’s instructions. Human-specific primers for URAT1 were designed using the Primer-BLAST software from the National Center for Biotechnology Information (NCBI), U.S. National Library of Medicine, as follows: forward primer 5′-CGGACCTGTATCTCCACGTT-3′, reverse primer 5′-TGCCTTCTTTACTGCCTGGT-3′, denaturation at 94 °C for 30 s, annealing at 60 °C for 45 s, elongation at 72 °C for 45 s, and 28 thermal cycles for a 570 base pair (bp) product length. The 18S-ribosomal RNA sequence was used as house-keeping gene control as follows: forward primer 5′-CGCGGTTCTATTTTGTTGGT-3′, reverse primer 5′-AGTCGGCATCGTTTATGGTC-3′, denaturation at 94 °C for 30 s, annealing at 60 °C for 30 s, elongation at 72 °C for 30 s, and 40 thermal cycles for a 570 bp product length. URAT1 mRNA expression was normalized using the house-keeping gene control and reported as 2^(∆∆Ct). The qPCR experiments are reported according to the Minimum Information for Publication of Quantitative Real-Time PCR Experiments (MIQE) guidelines to guarantee reproducibility.

### 2.6. Statistics

The Shapiro–Wilk test estimated the normality of data distribution. One-way ANOVA, followed by a post-hoc Tukey test, was used to compare the expression of CD11c, CD206, TNF-alpha, IL-1 beta, TLR4, CX3CR1, CCR2, and URAT1, as well as the intracellular amount of *E. coli*-GFP+ in monocyte-derived macrophages that were exposed to 0, 0.23, 0.45, or 0.9 mmol/L uric acid. All of the statistical analyses were performed using the GraphPad Prism 7 software. Differences were considered to be significant when *p* < 0.05.

## 3. Results

Upon differentiation in the presence of M-CSF for seven days, human monocyte-derived macrophages (MDM) showed significantly higher cell size than monocytes (Figure 1A,B). Furthermore, while monocytes exhibited a typical appearance that consisted of round cells, differentiated MDM appeared as fusiform-shaped cells with numerous pseudopodia that increased their cell complexity with respect to monocytes (Figure 1A,B). Additionally, macrophage differentiation was also confirmed by means of CD14 cell surface expression. In this sense, MDM showed a significant 20% reduction in CD14 expression when compared to that found in monocytes (Figure 1C). Once gated for singlets on a forward scatter height/forward scatter area density plot (Figure 1D), the living cells were gated while using the 7-AAD staining for dead cells. The living monocytes were then gated on a forward scatter area/side scatter area plot to assess CD14 positive expression together with TNF-alpha, IL-1 beta, CD11c, CD206, and CX3CR1, and CCR2 (Figure 1D).

Regarding the production of proinflammatory cytokines, the exposure of MDM to increasing uric acid concentrations tended to raise the number of TNF-alpha+ cells, although no significant changes were reached (Figure 2A). However, MDM in vitro cultured in the presence of 0.23 mmol/L uric acid showed a significant 25% increase in TNF-alpha production with respect to that found in the control cells (Figure 2B). No significant differences were found when compared MDM exposed to 0.45 or 0.9 mmol/L uric acid with control cells, thus suggesting that uric acid exerts its effects on TNF-alpha production in a dose-dependent fashion (Figure 2B). Despite exposure of MDM to increasing uric acid levels tended to diminish either the number of IL-1 beta+ cells or IL-1 beta production, no significant differences were found (Figure 2C,D, respectively).

We decided to measure TLR4 production as a key regulator of TNF-alpha synthesis since exposure of MDM to uric acid increased the expression of TNF-alpha, but not IL-1 beta. In this sense, uric acid induced TLR4 expression on MDM in a dose-dependent fashion, even though no significant differences were reached in terms of the number of TLR4+ cells (Figure 3A). However, exposure of MDM to 0.23 mmol/L uric acid promoted a significant 30% increase in TLR4 production in a very similar way than that observed for the case of TNF-alpha (Figure 3B). TNF-alpha positive macrophages have been also shown to produce CD11c, an integrin that is highly expressed in proinflammatory monocytes recruited toward atherosclerotic plaques. Uric acid did not affect the number of MDM expressing CD11c (Figure 3C); however, uric acid increased CD11c expression on MDM in a dose-dependent fashion, reaching the highest production of this integrin at 0.23 and 0.45 mmol/L uric acid (Figure 3D).

It is well known that macrophages gain proinflammatory ability while losing the anti-inflammatory capacity. For this reason, we wanted to measure CD206 production, a typical anti-inflammatory marker of human and murine macrophages. As expected, the exposure of MDM to increasing levels of uric acid progressively reduced the number of cells expressing CD206 with respect to control cells (Figure 4A). Overall, uric acid did not affect CD206 expression in MDM, except for 0.45 mmol/L uric acid that induced a slight increase in the expression of this anti-inflammatory marker (Figure 4B).

We decided to examine the effect of this metabolite on the expression of CX3CR1 and CCR2, two chemokine receptors that decreases in proinflammatory macrophages. The exposure of MDM to uric acid progressively diminished the number of CX3CR1+ macrophages, reaching the maximum response when cells were incubated in the presence of 0.9 mmol/L uric acid (Figure 5A). Likewise, uric acid gradually reduced CX3CR1 expression in MDM as compared to that found in the control cells (Figure 5B). In parallel, the exposure of MDM to uric acid gradually decreased the number of cells expressing CCR2, reaching the lowest level of this chemokine receptor when using 0.9 mmol/L uric acid (Figure 5C). As expected, uric acid also diminished CCR2 expression in MDM at 0.45 and 0.9 mmol/L and provided solid experimental evidence that CX3CR1 and CCR2 production behaves similarly in macrophages that were exposed either to LPS or uric acid.

Proinflammatory activation of human macrophages is also associated with an increased ability to phagocytose bacteria. Thus, we evaluated the effect of increasing uric acid concentrations on the phagocytic activity of MDM while using *E. coli*-GFP+. In the absence of uric acid, MDM were able to ingest around 15% of *E. coli*-GFP+ (Figure 6). Notably, the exposure of MDM to 0.23 mmol/L uric acid induced a significant 26% increase in the intracellular number of *E. coli*-GFP+ as compared to control cells (Figure 6). Accordingly, MDM exposed to 0.45 and 0.9 mmol/L uric acid showed significant 40 and 120% increases, respectively, in the intracellular amount of *E. coli*-GFP+ as compared to control cells (Figure 6), which confirmed that uric acid acts as a proinflammatory stimulus for in vitro cultured human macrophages.

An additional goal of this work was to explore the possible molecular mechanism by which uric acid exerts its effects on human monocyte-derived macrophages. For this reason, we decided to evaluate URAT1 expression, a key mediator of uric acid transport inside the cell. Notably, the exposure of MDM to 0.23 mmol/L uric acid induced a significant 90% reduction in URAT1 expression when compared to that found in control cells cultured in the absence of this metabolite (Figure 7). Similarly, MDM exposed to 0.45 mmol/L uric acid showed a significant 95% decrease in the mRNA levels of URAT1 with respect to that found in control cells (Figure 7). It is worth mentioning that exposure of MDM to 0.9 mmol/L uric acid totally abolished URAT1 expression (Figure 7), thus suggesting that URAT1 is sensitive to uric acid concentrations and it may take part in mediating the effects of this metabolite on macrophages.

We decided to use probenecid that is able to inhibit URAT1-dependent uric acid transport also examining its effect on TNF-alpha production and phagocytic activity to confirm the apparent involvement of URAT1 in macrophages. As expected, the use of probenecid totally abolished TNF-alpha production in macrophages that were also exposed to different concentrations of uric acid (Figure 8A). Likewise, probenecid also induced an 11% decrease in the intracellular number of *E. coli*-GFP+ macrophages that were treated with 0.23 mmol/L uric acid (Figure 8B). Furthermore, MDM treated with probenecid showed significant 78 and 61% reductions in the percentage of *E. coli*-GFP+, even in the presence of 0.45 and 0.9 mmol/L uric acid, respectively (Figure 8B). These results functionally suggest that uric acid might be transported inside the macrophage via URAT1 that, in turn, could mediate its proinflammatory effects on these immune cells.

## 4. Discussion

The association of hyperuricemia with the risk of developing metabolic abnormalities and cardiovascular diseases has been linked to an increased proinflammatory activity of macrophages [22,23]. However, it remains unknown whether hyperuricemia merely concurs with changes in macrophage activity or if uric acid is able to directly induce proinflammatory activation of human macrophages. For this reason, we decided to test the in vitro effect of increasing the uric acid concentrations on the proinflammatory profile of primary human macrophages differentiated from circulating monocytes.

We found that uric acid stimulates TNF-alpha production, but not IL-1 beta, in a dose-dependent fashion after confirming that human monocytes were properly differentiated into macrophages using a strategy that combined cell size and complexity as well CD14 expression. TNF-alpha and IL-1 beta are proinflammatory cytokines that play key roles in fever, cachexia, tumorigenesis inhibition, pyroptosis-related cell death, and immune cell recruitment [24,25,26]. However, TNF-alpha, but not IL-1 beta, has been consistently associated with increased serum levels of uric acid in several pathologic scenarios. For instance, the serum uric acid levels rise as TNF-alpha-producing monocytes also increase in women with preeclampsia [27]. Similarly, uric acid has been shown to stimulate TNF-alpha expression in vascular smooth muscle cells of Sprague–Dawley rats [28]. On the other hand, plasma IL-1 beta showed very poor association with increasing serum levels of uric acid in 1684 women and men, whereas TNF-alpha serum values rose in the same proportion than plasma uric acid [10]. Together with previous information, our results provide strong experimental evidence that uric acid might favor a proinflammatory state by predominantly stimulating TNF-alpha production in human macrophages. It is well known that TNF-alpha synthesis depends on the TLR4-dependent signaling pathway, whereas IL-1 beta production depends on the activity of the NLRP3 (NOD-LRR-and pyrin domain-containing protein 3) inflammasome [29,30,31]. For this reason, we decided to assess TLR4 in the same in vitro differentiated macrophages.

TLR4 is a transmembrane protein that is able to recognize numerous damage-associated molecular patterns (DAMP) and pathogen-associated molecular patterns (PAMPs), including free-fatty acids and LPS [32,33]. Upon activation, TLR4 is capable of inducing downstream nuclear factor-kappa B (NF-kB) activation, finally leading to TNF-alpha production [34]. Concurring with this information, our results demonstrate that TLR4 is produced in response to uric acid in the same way that TNF-alpha does in human macrophages. In this regard, a previous work reported that the risk of gout—a pathology known by deposition of monosodium urate crystals in joints—is directly associated with the polymorphism rs2149356 related to high TLR4 production [35]. Likewise, a very recent study demonstrated that uric acid promotes the mRNA expression of TLR4 in rat adipocytes in vitro [36]. Thus, we speculate that uric acid is able to induce TNF-alpha production via TLR4 activation, a phenomenon that provides proinflammatory features to human macrophages. However, the idea that uric acid can activate TLR4 should be taken with caution, since we only evaluated TLR4 at the protein level without assessing its ability as a cell signal transducer.

Until here, our results appeared to indicate that uric acid exerts the ability to polarize human macrophages towards a proinflammatory state. Thus, we decided to confirm the apparently proinflammatory capacity of macrophages by analyzing CD11c and CD206. CD11c is a beta-2 integrin that is highly expressed in monocytes and macrophages with prominent proinflammatory functions, whereas CD206—also referred to as the mannose receptor—is a C-type lectin that is predominantly expressed in murine and human macrophages exerting anti-inflammatory actions [37,38,39]. Interestingly, we found that uric acid is able to increase CD11c production at the same time, which reduces the number of macrophages expressing CD206. In this sense, it has been previously reported that blockage of uric acid synthesis by uricase treatment decreases the number of CD11c+ monocytes in mice [40], which suggested, for the first time, a relationship between uric acid and CD11c production. In parallel, a previous report showed that macrophages from synovial fluid of patients with gout tend to show reduced CD206 expression as compared to that found in the macrophages of patients with rheumatoid arthritis [41]. Altogether, this information concurs with the idea that macrophages adopt proinflammatory functions while losing anti-inflammatory capacities in the presence of elevated levels of uric acid.

The idea that uric acid might act as a direct proinflammatory stimulus for human macrophages is also supported by two additional facts: (a) the expression pattern of CX3CR1 and CCR2 and (b) the phagocytic activity of macrophages. CX3CR1 and CCR2 are both involved in mediating monocyte recruitment to the inflammation sites, where these cells will differentiate into macrophages and orchestrate inflammatory responses or tissue repair [42,43]. Interestingly, numerous studies have consistently reported the downregulation of CX3CR1 and CCR2 in the presence of prototypical inflammatory stimuli, such as LPS. In this sense, a seminal work reported that circulating monocytes either of septic patients or incubated with LPS show dramatically decreased CX3CR1 expression [44]. Similarly, in vitro and in vivo exposure of murine peripheral blood cells to LPS can downregulate CCR2 expression with direct consequences for macrophage migratory ability [45,46]. Therefore, CX3CR1 and CCR2 expression appears to behave similarly in macrophages that are exposed either to LPS or uric acid, which provides solid experimental evidence regarding the possible inflammatory role of this metabolite in macrophages. Additionally, it is well known that macrophages with proinflammatory functions show greater ability to phagocytose bacteria than that described in anti-inflammatory macrophages [19,20]. Interestingly, the exposure of macrophages to increasing uric acid concentrations progressively improved their phagocytic activity, which once again supports the notion that uric acid acts as a direct proinflammatory stimulus for these immune cells.

Besides studying the apparently proinflammatory effect of uric acid on macrophages, we also wanted to explore the molecular mechanism that is potentially involved. In our study, the production of TNF-alpha, TLR4, and CD11c in response to uric acid exhibited a typical dose-response relationship, where a maximum effect is found and, beyond this point, a plateau can be seen, which indicates saturation and the abolishment of the observed effect [47]. The dose-response relationship has been attributed to the interaction between ligand and its receptor [47,48], which suggested to us the possible involvement of a molecule able to transport uric acid inside the macrophage, as is the case of URAT1. In this way, we describe for, the first time, that human macrophages express URAT1, a transmembrane protein that had been only reported in endothelial cells, adipocytes, and cartilage chondrocytes [5]. Notably, we found that URAT1 expression in macrophages decreased as uric acid concentration increased, which might partially explain the fact that TNF-alpha, TLR4, and CD11c expression reached a saturation point, which in turn led to a decrease their protein levels. Taking into consideration that uric acid can increase NF-kB transcriptional activity in the pancreatic beta cell line Min6 [49], we thus speculate that TNF-alpha production in macrophages might depend on the interaction among uric acid, URAT1, and possibly NF-kB.

We performed functional assays aimed to pharmacologically block this urate transporter using probenecid to confirm the possible involvement of URAT1 in mediating the effects of uric acid on macrophages. Probenecid acts as a competitive inhibitor of URAT1 thus preventing reuptake and transport of uric acid by cells of the renal proximal tube [50]. Interestingly, the blockage of URAT1 abolished TNF-alpha production and phagocytic activity previously seen with uric acid, which suggests that the proinflammatory effect of uric acid entirely depends on its entry to macrophages. In this regard, it has been previously proposed that the entry of monosodium urate to THP-1 cells can induce IkB phosphorylation via Src family tyrosine kinases, thus leading to NF-kB activation and, finally, proinflammatory cytokine production [51]. In this way, our results confirm that (1) the entry of uric acid to macrophages has proinflammatry effects by a mechanism that is still unknown, and (2) entry of uric acid to macrophages seems to involve URAT1, whose expression is, in turn, sensitive to different concentrations of uric acid. However, URAT1 is not the only urate transporter and probenecid is not a URAT1 specific inhibitor, so measuring in macrophages other urate transporters, including organic anion transporter (OAT) 1, OAT3, and ATP-binding cassette subfamily G member 2 (ABCG2), and testing other urate uptake inhibitors, such as benzbromarone, dotinurad, and losartan, remains to be done.

## 5. Conclusions

Uric acid acts as a proinflammatory stimulus for in vitro cultured primary human macrophages through (a) increasing the production of TNF-alpha, TLR4, and CD11c, (b) improving the macrophage phagocytic activity, and (c) decreasing CD206, CX3CR1, and CCR2 expression. The possible mechanism by which uric acid exerts its proinflammatory effects on human macrophages appears to involve URAT1 in a dose-response fashion. URAT1 might, in turn, enhance NF-kB activation and lead to the production of proinflammatory cytokines by ways that remain to be elucidated. The use of probenecid functionally demonstrated that the entry of uric acid to the macrophage (d) has proinflammatory actions and (e) partially depends on URAT1. These results provide solid experimental evidence supporting the idea that elevated levels of uric acid can directly promote the macrophage-mediated systemic inflammatory state that is, in turn, associated with high cardiovascular risk in patients with chronic diseases. The idea that uric acid might act as a metabolic ligand with proinflammatory effects on human macrophages should be further examined.

## Figures and Tables

**Figure 1 biomolecules-10-00576-f001:**
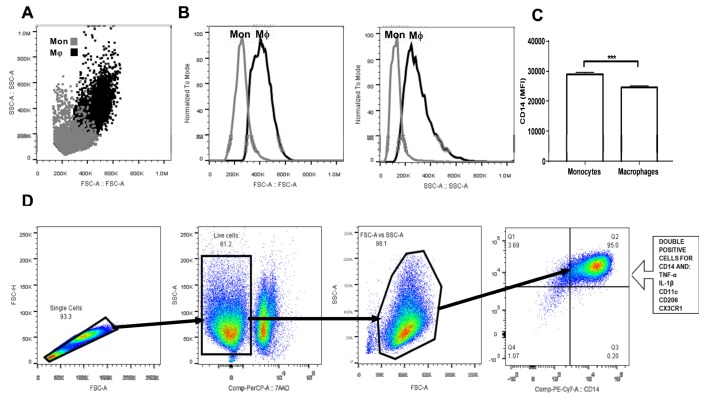
Gating strategy for primary human monocyte-derived macrophage characterization. Primary human monocytes were isolated from buffy coat suspensions (*n* = 10) and differentiated into macrophages in the presence of M-CSF for seven days. (**A**) Dot plot showing a comparison between monocytes (grey dots) and macrophages (black dots) in terms of cell size and complexity. (**B**) Macrophages are considerably larger and more complex than monocytes. (**C**) As previously reported, macrophages exhibit significantly decreased CD14 cell surface expression as compared to that found in monocytes. (**D**) Monocyte-derived macrophages (MDM) were gated for singlets on a forward scatter height/forward scatter area density plot. Resulting cells were gated again for detection of living macrophages by means of the 7AAD or Zombie UV stains. Macrophages were then gated on a side scatter area/forward scatter area density plot for detection of CD14+ cells that simultaneously expressed tumor necrosis factor-alpha (TNF-alpha), interleukin (IL-1) beta, CD11c, CD206, CX3CR1, and CCR2. Mon, monocyte population; Mϕ, macrophage population; MFI, mean fluorescence intensity; FSC-H, forward scatter height density plot; FSC-A, forward scatter area density plot; SSC-A, side scatter area density plot; TNF-a, tumor necrosis factor alpha; IL-1b, interleukin 1 beta; CD11c, cluster of differentiation 11c; CD206, cluster of differentiation 206 or mannose receptor; CX3CR1, CX3C-motif chemokine receptor 1; CCR2, C-C chemokine receptor type 2; M-CSF, macrophage-colony stimulating factor.

**Figure 2 biomolecules-10-00576-f002:**
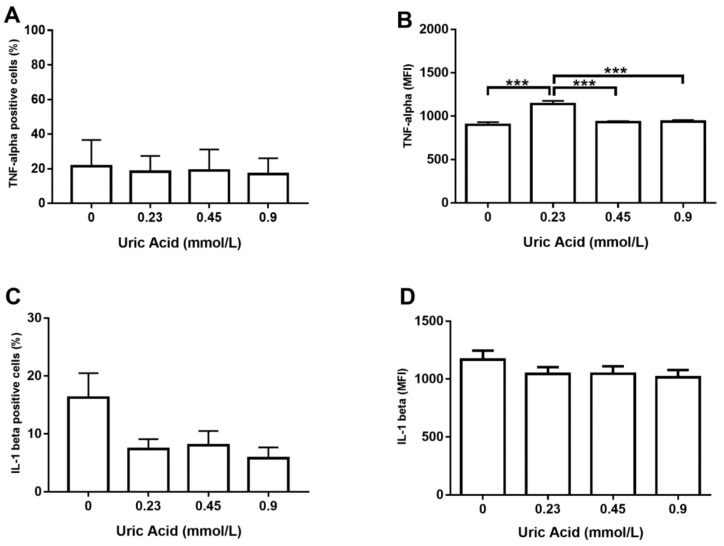
Intracellular production of TNF-alpha and IL-1 beta in primary human monocyte-derived macrophages in vitro exposed to uric acid. In vitro exposure to increasing concentrations of uric acid did not alter the percentage of primary human monocyte-derived macrophages expressing TNF-alpha (**A**). In vitro exposure to 0.23 mmol/L uric acid significantly increased TNF-alpha production in primary human monocyte-derived macrophages as compared to control cells (0 mmol/L uric acid) (**B**). In vitro exposure to increasing concentrations of uric acid tended to decrease the percentage of primary human monocyte-derived macrophages expressing IL-1 beta, although no significant differences were found (**C**). In vitro exposure to increasing concentrations of uric acid did not alter IL-1 beta production in primary human monocyte-derived macrophages (**D**). Data were analyzed using one-way ANOVA followed by the post-hoc Tukey test to estimate significant differences. Data are expressed as mean ± standard deviation. Differences were considered significant when *p* < 0.05 and are marked with asterisks as follows: *** = *p* < 0.0001. TNF-alpha, tumor necrosis factor alpha; IL-1 beta, interleukin 1 beta; MFI, mean fluorescence intensity.

**Figure 3 biomolecules-10-00576-f003:**
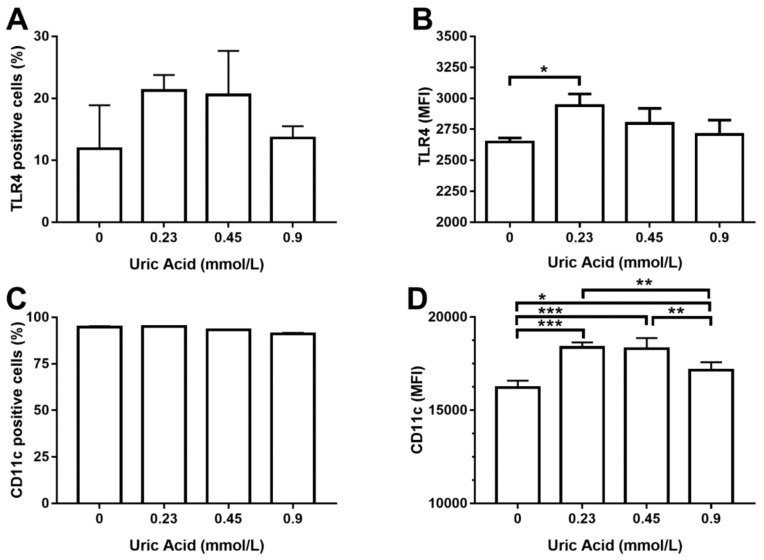
Cell surface expression of TLR4 and CD11c in primary human monocyte-derived macrophages in vitro exposed to uric acid. In vitro exposure to increasing concentrations of uric acid did not alter the percentage of primary human monocyte-derived macrophages expressing TLR4 (**A**). In vitro exposure to 0.23 mmol/L uric acid significantly increased TLR4 production in primary human monocyte-derived macrophages as compared to control cells (0 mmol/L uric acid) (**B**). In vitro exposure to increasing concentrations of uric acid did not alter the percentage of primary human monocyte-derived macrophages expressing CD11c (**C**). In vitro exposure to 0.23, 0.45, and 0.9 mmol/L uric acid significantly altered CD11c production in primary human monocyte-derived macrophages in a dose-response manner as compared to control cells (0 mmol/L uric acid) (**D**). Data were analyzed using one-way ANOVA, followed by the post-hoc Tukey test to estimate significant differences. Data are expressed as mean ± standard deviation. Differences were considered to be significant when *p* < 0.05 and are marked with asterisks as follows: * = *p* < 0.01; ** = *p* < 0.001; *** = *p* < 0.0001. TLR4, toll-like receptor 4; CD11c, cluster of differentiation 11c; MFI, mean fluorescence intensity.

**Figure 4 biomolecules-10-00576-f004:**
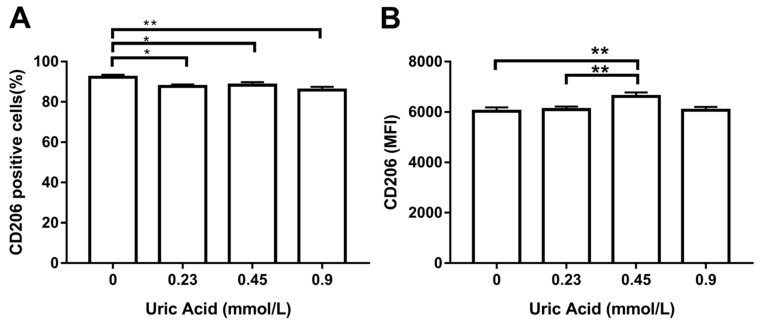
Cell surface expression of CD206 in primary human monocyte-derived macrophages in vitro exposed to uric acid. In vitro exposure to 0.23, 0.45, and 0.9 mmol/L uric acid significantly decreased the percentage of primary human monocyte-derived macrophages expressing CD206 as compared to control cells (0 mmol/L uric acid) (**A**). In vitro exposure to 0.45 mmol/L uric acid significantly increased CD206 production in primary human monocyte-derived macrophages as compared to control cells (0 mmol/L uric acid) (**B**). Data were analyzed using one-way ANOVA followed by the post-hoc Tukey test to estimate significant differences. Data are expressed as mean ± standard deviation. Differences were considered significant when *p* < 0.05 and are marked with asterisks as follows: * = *p* < 0.01; ** = *p* < 0.001. CD206, cluster of differentiation 206 or mannose receptor; MFI, mean fluorescence intensity.

**Figure 5 biomolecules-10-00576-f005:**
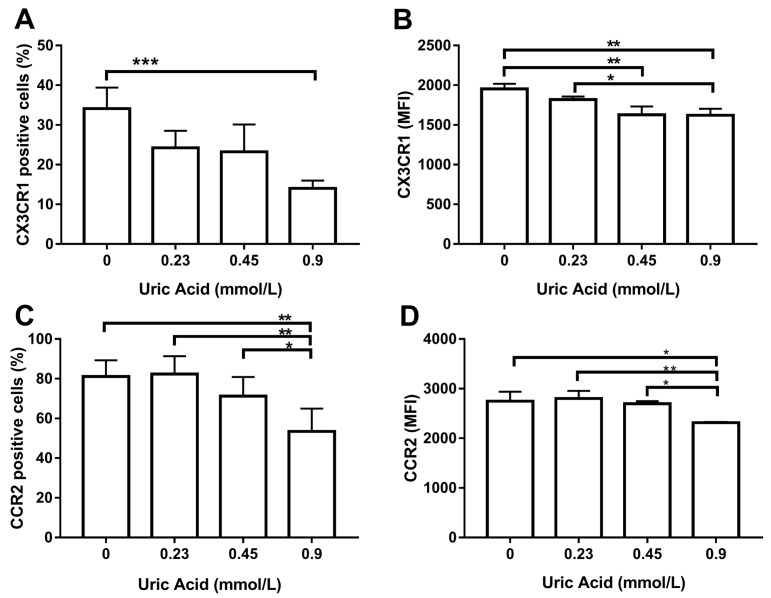
Cell surface expression of CX3CR1 and CCR2 in primary human monocyte-derived macrophages in vitro exposed to uric acid. In vitro exposure to 0.9 mmol/L uric acid significantly decreased the percentage of primary human monocyte-derived macrophages expressing CX3CR1 as compared to control cells (0 mmol/L uric acid) (**A**). In vitro exposure to 0.45 and 0.9 mmol/L uric acid significantly decreased CX3CR1 production in primary human monocyte-derived macrophages as compared to control cells (0 mmol/L uric acid) (**B**). In vitro exposure to 0.9 mmol/L uric acid significantly decreased the percentage of primary human monocyte-derived macrophages expressing CCR2 as compared to control cells (0 mmol/L uric acid (**C**). In vitro exposure to 0.9 mmol/L uric acid significantly decreased CCR2 production in primary human monocyte-derived macrophages as compared to control cells (0 mmol/L uric acid) (**D**). Data were analyzed using one-way ANOVA followed by the post-hoc Tukey test to estimate significant differences. Data are expressed as mean ± standard deviation. Differences were considered to be significant when *p* < 0.05 and are marked with asterisks as follows: * = *p* < 0.01; ** = *p* < 0.001; *** = *p* < 0.0001. CX3CR1, CX3C-motif chemokine receptor 1; CCR2, C-C chemokine receptor type 2; MFI, mean fluorescence intensity.

**Figure 6 biomolecules-10-00576-f006:**
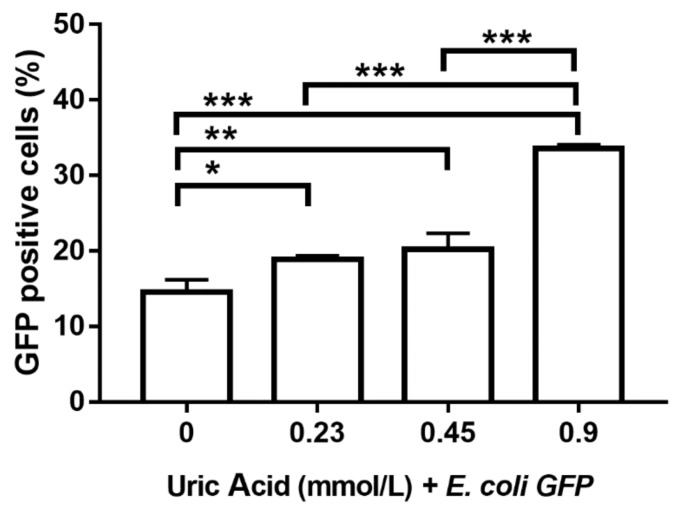
Bacterial phagocytic activity of primary human monocyte-derived macrophages in vitro exposed to uric acid. In vitro exposure to 0.23, 0.45, and 0.9 mmol/L uric acid significantly enhanced the phagocytic activity of primary human monocyte-derived macrophages by progressively increasing the percentage of *Escherichia coli*-green fluorescent protein (GFP) positive cells as compared to controls (0 mmol/L uric acid). Data were analyzed using one-way ANOVA followed by the post-hoc Tukey test to estimate significant differences. Data are expressed as mean ± standard deviation. Differences were considered significant when *p* < 0.05 and are marked with asterisks as follows: * = *p* < 0.01; ** = *p* < 0.001; *** = *p* < 0.0001. GFP, green fluorescent protein; *E. coli*, *Escherichia coli*.

**Figure 7 biomolecules-10-00576-f007:**
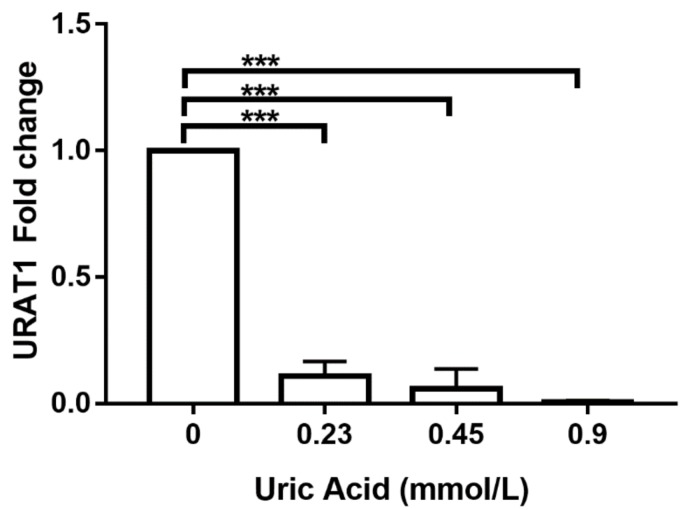
URAT1 expression in primary human monocyte-derived macrophages in vitro exposed to uric acid. In vitro exposure to 0.23, 0.45, and 0.9 mmol/L uric acid progressively abolished URAT1 expression in primary human monocyte-derived macrophages as compared to controls (0 mmol/L uric acid). URAT1 expression was normalized using the 18S-ribosomal RNA gene as house-keeping control gene and reported as 2^(∆∆Ct)-fold change by real-time polymerase chain reaction (qPCR) using SYBR Green Master Mix and AmpliTaq^®^ Fast DNA Polymerase. Data were analyzed using one-way ANOVA followed by the post-hoc Tukey test to estimate significant differences. Data are expressed as mean ± standard deviation. Differences were considered significant when *p* < 0.05 and are marked with asterisks as follows: *** = *p* < 0.0001. URAT1, urate anion transporter 1.

**Figure 8 biomolecules-10-00576-f008:**
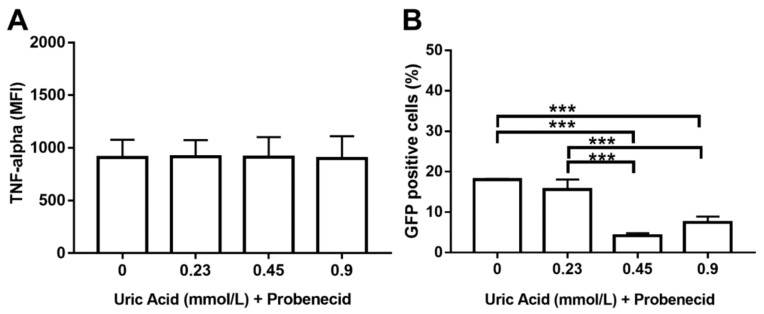
Probenecid seems to abolish URAT1-dependent proinflammatory effects of uric acid on primary human monocyte-derived macrophages. In vitro exposure to 1 mmol/L probenecid abolished the stimulatory effect of uric acid on TNF-alpha production in primary human monocyte-derived macrophages as compared to cells exposed to uric acid in the absence of probenecid (**A**). In vitro exposure to 1 mmol/L probenecid significantly decreased the stimulatory effect of uric acid on the phagocytic activity of primary human monocyte-derived macrophages by decreasing the percentage of Escherichia coli-GFP positive cells as compared to macrophages exposed to uric acid in the absence of probenecid (**B**). Data were analyzed using one-way ANOVA followed by the post-hoc Tukey test to estimate significant differences. Data are expressed as mean ± standard deviation. Differences were considered significant when *p* < 0.05 and are marked with asterisks, as follows: *** = *p* < 0.0001. TNF-a, tumor necrosis factor alpha; MFI, mean fluorescence intensity; GFP, green fluorescent protein.

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
