# Peer review of "Uric Acid Has Direct Proinflammatory Effects on Human Macrophages by Increasing Proinflammatory Mediators and Bacterial Phagocytosis Probably via URAT1"

_biomolecules, 2020, doi:10.3390/biom10040576_

Round 1
Reviewer 1 Report
Martínez-Reyes et. al. demonstrated that urate made human macrophage express proinflammatory markers, which was inhibited by probenecid treatment. The author advocated that URAT1 should be related to this proinflammatory effect of urate. However, there are serious logical gaps which make this manuscript be revised.
Major point
The qPCR data means human macrophages express URAT1 mRNA. However, there is possibility of the expression of other urate transporters, GLUT9, ABCG2, OAT1, OAT3, OAT4, OAT10 in human macrophage. Authors must decide the expressions of these other urate transporter genes in human macrophage.
Because probenecid is non-specific inhibitor of organic anion transporters, the inhibition of urate effect by probenecid does not mean the inhibition urate transport via URAT1. If human macrophage express OAT1, probenecid will inhibit urate uptake via OAT1. To demonstrate the inhibition of urate uptake via URAT1, URAT1-specific inhibitors, benzbromarone or dotinurad, must be used.
Minor point
Page 1 line 41 “hepatocyte”
There is no report which demonstrate that hepatocytes express URAT1 in ref 2-5.
Page 4 line 149 “Mouse specific primers”
These primers are not mouse specific but human specific.
Author Response
Martínez-Reyes et. al. demonstrated that urate made human macrophage express proinflammatory markers, which was inhibited by probenecid treatment. The author advocated that URAT1 should be related to this proinflammatory effect of urate. However, there are serious logical gaps which make this manuscript be revised.
Reply (R)
We thank to the Reviewer for her/his positive feedback regarding our work. We tried to resolve all gaps you mention and think the last version of this work has been improved.
Query (Q) 1
Major point
The qPCR data means human macrophages express URAT1 mRNA. However, there is possibility of the expression of other urate transporters, GLUT9, ABCG2, OAT1, OAT3, OAT4, OAT10 in human macrophage. Authors must decide the expressions of these other urate transporter genes in human macrophage.
R1
URAT1 is considered the major urate reabsorption transporter in the apical membrane of proximal tubule epithelial cells (Xu L et al., 2017, Oncotarget 10;8(59):100852-100862). For this reason, we decided to investigate whether human macrophages expressed URAT1 before considering measuring other urate reabsorption transporters such as OAT4 and GLUT9. As we indeed detected URAT1 mRNA expression on in vitro culture human macrophages, we decided to go that direction in this first work. However, the Reviewer is totally right, and we are now working on measuring not only other urate reabsorption transporters, but also urate excretion transporters including OAT1, OAT3, ABCG2, and MRP4. Nevertheless, please consider that such a work will take a couple years before we can categorically demonstrate what kind of uric acid transporters are expressed by human macrophages. In that case, data resulting from that new research line will be published other time.
Q2
Because probenecid is non-specific inhibitor of organic anion transporters, the inhibition of urate effect by probenecid does not mean the inhibition urate transport via URAT1. If human macrophage express OAT1, probenecid will inhibit urate uptake via OAT1. To demonstrate the inhibition of urate uptake via URAT1, URAT1-specific inhibitors, benzbromarone or dotinurad, must be used.
R2
Besides having been used for decades for treating patients with gout, we decided to use probenecid because it is considered a high affinity URAT1 inhibitor. However, the Reviewer is right, and probenecid is not specific URAT1 inhibitor. Nevertheless, we respectfully want to note that benzbromarone is also not a specific URAT1 inhibitor. In 2009, Bibert and coworkers demonstrated that benzbromarone inhibits urate uptake via GLUT9 in Xenopus laevis oocytes (Bibert S et al., 2009, Am J Physiol Renal Physiol 297(3):F612-9). In the same sense, dotinurad is a very recent selective urate reabsorption inhibitor developed in Japan that blocks urate uptake via URAT1, OAT1, and OAT3 in MDCKII cells (Taniguchi T et al., 2019, J Pharmacol Exp Ther 371(1):162-170). However, we totally agree with the Reviewer’s observation and besides starting new in vitro experiments to test the ability of benzbromarone and dotinurad to inhibit urate uptake in human macrophages (results that will be published further), we have decided to modify the article title, abstract, introduction, material and methods, results, discussion, and conclusions, with the aim of stating there are additional urate transporters and URAT1 inhibitors that might mediate the effect of uric acid on human macrophages. Please see the abovementioned changes marked with yellow color at lines 4, 32, 34, 35, 41, 101, 334, 338, 444-449, 455, and 458. Thanks for your very positive feedback; we sincerely think your comments helped us to improve the last version of this manuscript.
Q3
Minor point
Page 1 line 41 “hepatocyte”
There is no report which demonstrate that hepatocytes express URAT1 in ref 2-5.
R3
We apologize for this terrible mistake. We wrongly written hepatocytes instead of chondrocytes. We have formally requested that current #5 reference be replaced by Zhang B, Duan M, Long B, Zhang B, Wang D, Zhang Y, Chen J, Huang X, Jiao Y, Zhu L, Zeng X. Urate transport capacity of glucose transporter 9 and urate transporter 1 in cartilage chondrocytes. Mol Med Rep. 2019;20(2):1645-1654. Please see this change marked with green color at page 1, line 42, and at page 14, line 472, and page 15, lines 484-487. Thanks for your accurate observation.
Q4
Page 4 line 149 “Mouse specific primers”
These primers are not mouse specific but human specific.
R4
We apologize for this terrible mistake. It has been corrected as requested. Please see this change marked with yellow color at page 4 line 150.
Reviewer 2 Report
Dr Martinez-Reyes et al, reported the inflammatory effects of uric acid in human macrophages, focusing on the evaluation of proinflammatory molecules and bacterial phagocytic activity. Then, they suggested that uric acid could exert its proinflammatory action by URAT1 and to this aim, they blocked it by probenecid. This result is interesting because, for the first time, URAT1 expression is found in human macrophages.
Therefore, I have some questions:
1) Macrophages were exposed for 12 hours to 0.23-0.9 mmol/l uric acid, but only the lower concentration (a normal level) increased TNFα and TLR4, whereas 0.45-0.9 mmol/l (values associated with hyperuricemia) had no effects on their expression, but instead they had a dose dependent action on bacterial phagocytic activity of macrophages. I was wondering if the 12 hour exposition could be too long. Probably 0.45-0.9 mmol/L are very potent inflammatory triggers and then, a shorter incubation time needs. Did you perform any time course experiments? If not, it could be desirable to test whether in macrophages, hyperuricemia induces a strong and early inflammatory response. This information could highlight more the phagocytic activity induced by 0.45-0.9 mmol/L concentration.
2) The authors should evaluate NF-kB activation by uric acid.
3) URAT1 expression in human macrophages is the most important data in this manuscript, but some improvements need. The authors studied only URAT1 mRNA expression, but protein expression is necessary. Possibly, it could be detected by immunofluorecence and western blot. this information is too important for the strenght of their research.
4) The uric acid internalization is blocked by probenecid, that inhibits not only URAT1, but also other OATs. To verify that URAT1 engagement is implicated in macrophage inflammatory phenotype, I'd suggest:
Losartan treatment macrophages. Losartan is a well studied URAT1 inhibitor (AJH 2008, 10 1157-1162)
URAT1 knock-down by specific si-RNA
Author Response
Dr Martinez-Reyes et al, reported the inflammatory effects of uric acid in human macrophages, focusing on the evaluation of proinflammatory molecules and bacterial phagocytic activity. Then, they suggested that uric acid could exert its proinflammatory action by URAT1 and to this aim, they blocked it by probenecid. This result is interesting because, for the first time, URAT1 expression is found in human macrophages.
Therefore, I have some questions:
Reply (R)
We want to give special thanks to the Reviewer for her/his very constructive comments regarding our work.
Question (Q) 1
Macrophages were exposed for 12 hours to 0.23-0.9 mmol/l uric acid, but only the lower concentration (a normal level) increased TNFα and TLR4, whereas 0.45-0.9 mmol/l (values associated with hyperuricemia) had no effects on their expression, but instead they had a dose dependent action on bacterial phagocytic activity of macrophages. I was wondering if the 12 hour exposition could be too long. Probably 0.45-0.9 mmol/L are very potent inflammatory triggers and then, a shorter incubation time needs. Did you perform any time course experiments? If not, it could be desirable to test whether in macrophages, hyperuricemia induces a strong and early inflammatory response. This information could highlight more the phagocytic activity induced by 0.45-0.9 mmol/L concentration.
R1
We already performed several dose-time-response curves at 1, 3, 6, and 12 hours, and 1, 3, 6 and 9 days, and found that human monocyte-derived macrophages show the most significant increase in TNF-alpha, TLR4, and bacterial phagocytic activity after 12 hours on in vitro culture conditions in the presence of uric acid. Thus, we think uric acid has indeed proinflammatory actions on human macrophages but not so impressive as compared to those found with other prototypical inflammatory stimuli such as LPS. We originally considered not important to include such information and decided to present only the most significant changes at 12 hours. However, in view of the Reviewer’s observation, we have decided to make a short mention at the Material and Method section stating why we chose 12 hours as optimal time for macrophage exposition to 0.23, 0.45, and 0.9 mmol/L uric acid. Please see this change marked with green color at page 3, lines 101-104.
Q2
The authors should evaluate NF-kB activation by uric acid.
R2
As the Reviewer suggest, we will definitely assess uric acid-dependent NF-kappa B activation in human macrophages. In fact, the main findings reported in this study have led us to speculate that somehow uric acid can induce IKK phosphorylation and then NF-kappa B activation (as it has been demonstrated in the pancreatic beta cell line Min6), which in turn leads to TNF-alpha production (please see this hypothesis marked with green color at pages 13 and 14, lines 428-436). However, we kindly ask the Reviewer to consider that characterization of the NF-kappa B-dependent signaling pathway that orchestrates the effects of uric acid on macrophages involves a whole experimental design that will take several months of work and be published in a subsequent research manuscript. Nevertheless, we want to thank to the Reviewer for her/his constructive ideas that have indubitably strengthened our research line.
Q3
URAT1 expression in human macrophages is the most important data in this manuscript, but some improvements need. The authors studied only URAT1 mRNA expression, but protein expression is necessary. Possibly, it could be detected by immunofluorecence and western blot. this information is too important for the strenght of their research.
R3
In this first work, we decided to determine whether human macrophages could express URAT1 mRNA, also examining whether URAT1 gene expression could be regulated by uric acid as its main ligand. Now we know URAT1 gene expression is potentially involved in mediating the proinflammatory effects of uric acid on macrophages, our next step is to assess URAT1 protein by flow cytometry and immunofluorescence. However, we firstly need to couple human anti-URAT1 primary antibodies to FITC or Red Texas, because there are not commercial fluorescent antibodies available in the market. This is the work we have been doing during the last 6 months. In parallel, we will take your advice on detecting URAT1 protein from human macrophages by western blot. However, we consider that these experiments are essential in a second part of this study, where we will characterize the whole profile of urate transporters in primary human macrophages. Thanks for giving such a positive feedback on this regard.
Q4
The uric acid internalization is blocked by probenecid, that inhibits not only URAT1, but also other OATs. To verify that URAT1 engagement is implicated in macrophage inflammatory phenotype, I'd suggest:
Losartan treatment macrophages. Losartan is a well studied URAT1 inhibitor (AJH 2008, 10 1157-1162).
URAT1 knock-down by specific si-RNA
R4
It is a remarkable suggestion. Besides having been used for decades for treating patients with gout, we decided to use probenecid because it is considered a high affinity URAT1 inhibitor. However, the Reviewer is right, and probenecid is not specific URAT1 inhibitor. Nevertheless, we respectfully want to note that losartan is also not a specific URAT1 inhibitor. In 2009, Bibert and coworkers demonstrated that 1mM losartan inhibits urate uptake via GLUT9 in Xenopus laevis oocytes (Bibert S et al., 2009, Am J Physiol Renal Physiol 297(3):F612-9). In the same sense, Sato and coworkers previously reported that 10 mM losartan inhibits urate uptake via MRP4 in Sf9 cells (Sato M et al., 2008, Pharm Res 25(3):639-46). However, we totally agree with the Reviewer’s observation and besides starting new in vitro experiments to test the ability of losartan to inhibit urate uptake in human macrophages (results that will be published further), we have decided to modify the article title, abstract, introduction, material and methods, results, discussion, and conclusions, with the aim of stating there are additional urate transporters and URAT1 inhibitors that might mediate the effect of uric acid on human macrophages. Please see the abovementioned changes marked with yellow color at lines 4, 32, 34, 35, 41, 101, 334, 338, 444-449, 455, and 458.
In parallel, URAT1 knock-down by specific si-RNA is also an extraordinary suggestion to start a totally new experimental series. However, please consider that such a work will take at least a year before knowing the percentage of urate transported by URAT1 in macrophages, and the possible involvement of others uric acid transporters expressed in the absence of URAT1 such as OAT1, OAT3, OAT4, ABCG2, GLUT9, and MRP4. In that case, data resulting from that new research line will be published other time.
Thank you so much for your very positive feedback; we think your comments helped us to improve the last version of this manuscript.
Round 2
Reviewer 1 Report
Martinez-Reyes et. al. respond appropriately to my request of revision. I agree this manuscript to be accepted.
Reviewer 2 Report
The authors mostly addressed my concerns.